# Improving Generalizability of Protein Sequence Models With Data Augmentations

## Abstract

While protein sequence data is an emerging application domain for machine learning methods, small modifications to protein sequences can result in difficult-to-predict changes to the protein's function. Consequently, protein machine learning models typically do not use randomized data augmentation procedures analogous to those used in computer vision or natural language, e.g., cropping or synonym substitution. In this paper, we empirically explore a set of simple string manipulations, which we use to augment protein sequence data when fine-tuning semi-supervised protein models. We provide 276 different comparisons to the Tasks Assessing Protein Embeddings (TAPE) baseline models, with Transformer-based models and training datasets that vary from the baseline methods only in the data augmentations and representation learning procedure. For each TAPE validation task, we demonstrate improvements to the baseline scores when the learned protein representation is fixed between tasks. We also show that contrastive learning fine-tuning methods typically outperform masked-token prediction in these models, with increasing amounts of data augmentation generally improving performance for contrastive learning protein methods. We find the most consistent results across TAPE tasks when using domain-motivated transformations, such as amino acid replacement, as well as restricting the Transformer attention to randomly sampled sub-regions of the protein sequence. In rarer cases, we even find that information-destroying augmentations, such as randomly shuffling entire protein sequences, can improve downstream performance.

## 1 Introduction

Semi-supervised learning has proven to be an effective mechanism to promote generalizability for protein machine learning models, as task-specific labels are generally very sparse. However, with other common data types there are simple transformations that can be applied to the data in order to improve a model's ability to generalize: for instance, vision models use cropping, rotations, or color distortion; natural language models can employ synonym substitution; and time series data models benefit from window restriction or noise injection. Scientific data, such as a corpus of protein sequences, have few obvious transformations that can be made to it that unambiguously preserve the meaningful information in the data. Often, an easily understood transformation to a protein sequence (*e.g.*, replacing an amino acid with a chemically similar one) will unpredictably produce either a very biologically similar or very biologically different mutant protein.

In this paper, we take the uncertainty arising from the unknown effect of simple data augmentations in protein sequence modeling as an empirical challenge that deserves a robust assessment. To our knowledge, no study has been performed to find out whether simple data augmentation techniques improve a suite of protein tasks. We focus on fine-tuning previously published self-supervised models that are typically used for representation learning with protein sequences, *viz.* the transformer-based methods of Rao et al. (2019) which have shown the best ability to generalize on a set of biological tasks, which are referred to as Tasks Assessing Protein Embeddings (TAPE). We test one or more of the following data augmentations: replacing an amino acid with a pre-defined alternative; shuffling the input sequences either globally or locally; reversing the sequence; or subsampling the sequence to focus only on a local region (see Fig. 1).

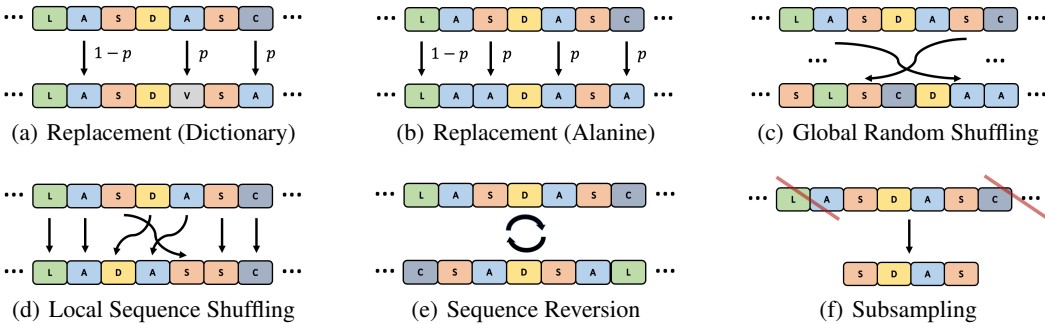

Figure 1: Diagram of data augmentations. We study randomly replacing residues (with probability $p$) with (a) a chemically-motivated dictionary replacement or (b) the single amino acid alanine. We also consider randomly shuffling either (c) the entire sequence or (d) a local region only. Finally, we look at (e) reversing the whole sequence and (f) subsampling to a subset of the original.

We demonstrate that the protein sequence representations learned by fine-tuning the baseline models with data augmentations results in relative improvements between 1% (secondary structure accuracy) and 41% (fluorescence $\rho$), as assessed with linear evaluation for all TAPE tasks we studied. When fine-tuning the same representations during supervised learning on each TAPE task, we show significant improvement as compared to baseline for 3 out of 4 TAPE tasks, with the fourth (fluorescence) within $1\sigma$ in performance. We also study the effect of increasingly aggressive data augmentations: when fine-tuning baseline models with contrastive learning (Hadsell et al., 2006; Chen et al., 2020a) we see a local maximum in downstream performance as a function of the quantity of data augmentation, with "no augmentations" generally under-performing modest amounts of data augmentations. Conversely, performing the same experiments but using masked-token prediction instead of contrastive learning, we detect a minor trend of decreasing performance on the TAPE tasks as we more frequently use data augmentations during fine-tuning. We interpret this as evidence that contrastive learning techniques, which require the use of data augmentation, are important methods that can be used to improve generalizibility of protein models.

## 2 RELATED WORKS

Self-supervised and semi-supervised methods have become the dominant paradigm in modeling protein sequences for use in downstream tasks. Rao et al. (2019) have studied next-token and masked-token prediction, inspired by the BERT natural language model (Devlin et al., 2018). Riesselman et al. (2019) have extended this to autoregressive likelihoods; and Rives et al. (2019), Heinzinger et al. (2019) and Alley et al. (2019) have shown that unsupervised methods trained on unlabeled sequences are competitive with mutation effect predictors using evolutionary features.Of importance to this work are self-supervised learning algorithms employed for other data types that use or learn data augmentations. For example, Gidaris et al. (2018) learn image features through random rotations; Dosovitskiy et al. (2014) and Noroozi & Favaro (2016) study image patches and their correlations to the original samples. van den Oord et al. (2018) uses contrastive methods to predicts future values of an input sequence. We consider sequence augmentations in natural language as the most relevant comparison for the data augmentations we study in this paper. Some commonly applied augmentations on strings include Lexical Substitution (Zhang et al., 2015), Back Translation (Xie et al., 2019a), Text Surface TransformationPermalink (Coulombe, 2018), Random Noise Injection (Xie et al., 2019b; Wei & Zou, 2019), and Synonym Replacement, Random Swap, Random Deletion (RD) (Wei & Zou, 2019). However, sequence augmentations designed for natural languages often require the preservation of contextual meaning of the sentences, a factor that is less explicit for protein sequences.

Contrastive Learning is a set of approaches that learn representations of data by distinguishing positive data pairs from negative pairs (Hadsell et al., 2006). SimCLR (v1 & v2) (Chen et al., 2020a;b) describes the current state-of-the-art contrastive learning technique.; we use this approach liberally in this paper not only because it performs well, but because it requires data transformations to ex-

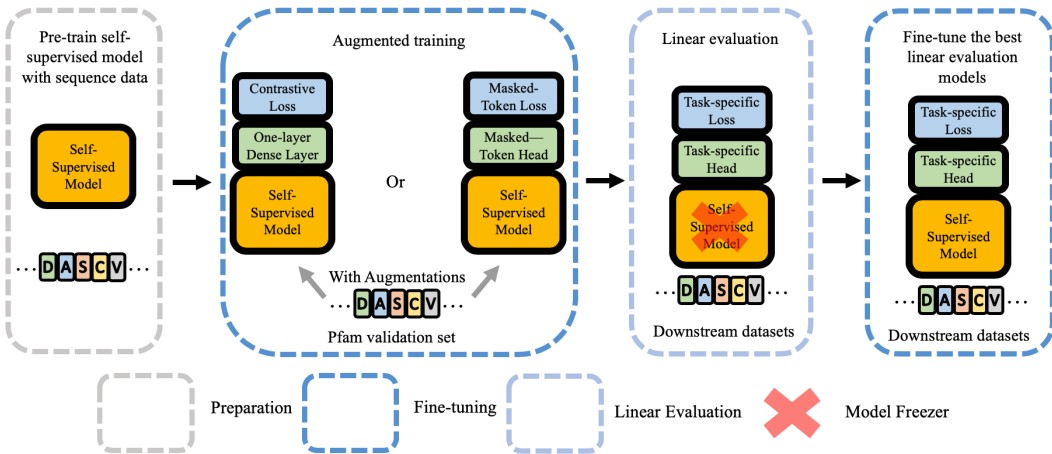

Figure 2: Diagram of experimental approach (see Sect. 3.1). We use dashed boxes to indicate different steps: semi-supervised pre-training, augmented learning, linear evaluation, and finally fine-tuning the best performing augmented models on downstream tasks. In each box, we include the general model architectures, with major sub-modules in different colors. The model freezer indicates the semi-supervised model is not updated during linear evaluation;.

ist. Since we focus on protein sequence transformations, the contrastive learning part described in both (Chen et al., 2020a;b) is our focus. Following Chen et al. (2020a;b), we denote input data as $x \in \mathbb{D}$, with $\mathbb{D}$ being our training set; we then define an embedding function, $f_\omega : x \mapsto h$ with $h \in \mathbb{R}^N$, and a mapping function $g_\theta : h \mapsto z$, with $z \in \mathbb{R}^M$, where $\omega$ and $\theta$ are the learned model weights. For any $x$, we form two copies $x_1 = t_1(x)$ and $x_2 = t_2(x)$ given functions $t_1, t_2 \sim T$, where $T$ denotes the distribution of the augmentation functions. Given $\mathbb{D}$ is of size $N$, the contrastive loss is written as:

$$L = \frac{1}{2N} \sum_{k=1}^{N} [l(z_k^{(1)}, z_k^{(2)}) + l(z_k^{(2)}, z_k^{(1)})] \quad \text{where} \quad l(u, v) \equiv -\log \frac{e^{\text{sim}(u,v)/\tau}}{\sum_{w \neq u} e^{\text{sim}(u,w)/\tau}} \quad (1)$$

Here, $z_{i,k} = g_\theta(f_\omega(t_i(x_k)))$, $\text{sim}(\cdot, \cdot)$ is cosine similarity, and $\tau \in (0, \infty)$ is a scalar temperature; we choose $\tau = 0.2$. By minimizing the contrastive loss, we obtain the learned $h$ as the encoded feature for other downstream tasks. Note, the contrastive loss takes $z$'s as inputs, whereas the encoded feature is $h$, which is the variable after the function $f_\omega(\cdot)$ and before $g_\theta(\cdot)$.

## 3 METHOD

### 3.1 EVALUATION PROCEDURE & APPROACH TO EXPERIMENT CONTROL

Our goal is to demonstrate that training self-supervised protein sequence models, with simple string manipulations as data augmentations, will lead to better performance on downstream tasks. To attempt to control external variables, we study the following restricted setting; we provide the procedural diagram in Figure 2 and the corresponding explanations of the four major steps below (See Appendix A for training setups in details.):

**Baseline.—** A self-supervised model $M_0$ is trained on non-augmented sequence data $D_{\text{seq}}$ to do representation learning. To have a consistent baseline, we set $M_0$ to the Transformer-based model trained and published in Rao et al. (2019), without modification. This was trained with masked-token prediction on Pfam protein sequence data (El-Gebali et al., 2019); it has 12 self-attention layers, 8 heads per layer, and 512 hidden dimensions, yielding 38M parameters in total.

**Augmented training on validation set.—** We fine-tune $M_0$ on augmented subsets $D_{\text{val}} \subset D_{\text{seq}}$, given a set of pre-defined data transformations $T_{\text{aug}}$. We define $M_{\text{aug}}$ as the final trained model derived from $T_{\text{aug}}(D_{\text{seq}})$ with $M_0$ as the initial conditions for the model parameters. We explore two different methods of fine-tuning on augmented data — a contrastive task (as in Eq. 1) and a

masked-token task (exponentiated cross entropy loss) — as well as different combinations of data augmentations. We use reduced subsets $|D_{\mathrm{val}}| \ll |D_{\mathrm{seq}}|$ to both reduce the computational cost of running bulk experiments, as well as to protect against overfitting. For consistency, we inherit the choice of $D_{\mathrm{val}}$ from the cross-validation split used to train $M_0$ in Rao et al. (2019). To adapt the same baseline model $M_0$ to different self-supervised losses, we add a loss-specific randomly-initialized layer to the $M_0$ architecture: contrastive learning with a fully connected layer that outputs 256 dimensional vectors and masked-token uses fully connected layers with layer normalization to output one-hot vectors for each of the masked letters. We define our different choices of $T_{\mathrm{aug}}$ in the next section.

**Linear evaluation on TAPE.—** To assess the representations learned by $M_{\mathrm{aug}}$, we evaluate performance on four TAPE downstream training tasks (Rao et al., 2019): stability, fluorescence, remote homology, and secondary structure. For consistency, we use the same training, validation, and testing sets. The first two tasks are evaluated by Spearman correlation ($\rho$) to the ground truth and the latter two by classification accuracy. However, we do not consider the fifth TAPE task, contact map prediction, as this relies only on the single CASP12 dataset, which has an incomplete test set due to data embargoes (AlQuraishi, 2019). Secondary structure prediction is a sequence-to-sequence task where each input amino acid is classified to a particular secondary structure type (helix, beta sheet, or loop), which is evaluated on data from CASP12, TS115, and CB513 (Berman et al., 2000; Moult et al., 2018; Klausen et al., 2019), with specifically "3-class" classification accuracy being the metric in this paper. The remote homology task classifies sequences into one of 1195 classes, representing different possible protein folds, which are further grouped hierarchically into families, then superfamilies; the datasets are derived from Fox et al. (2013). The fluorescence task regresses a protein sequence to a real-valued log-fluorescence intensity measured in Sarkisyan et al. (2016). The stability task regresses a sequence to a real-valued measure of the protein maintaining its fold above a concentration threshold (Rocklin et al., 2017). We perform linear evaluation by training only a single linear layer for each downstream task for each contrastive-learning model $M_{\mathrm{aug}}$, but not changing the parameters of $M_{\mathrm{aug}}$, and its corresponding learned encodings, across all tasks. To compare the contrastive learning techniques to further fine-tuning with masked-token prediction, we identify the best-performing data augmentations per-task, then replace the $M_{\mathrm{aug}}^{\mathrm{CL}}$ with the masked-token model with the same augmentation $M_{\mathrm{aug}}^{\mathrm{MT}}$, then also do linear evaluation on $M_{\mathrm{aug}}^{\mathrm{MT}}$.

**Full fine-tuning on TAPE.—** For the best-performing augmented models in the linear evaluation task (either $M_{\mathrm{aug}}^{\mathrm{CL}}$ or $M_{\mathrm{aug}}^{\mathrm{MT}}$), we further study how the models improve when allowing the parameters of $M_{\mathrm{aug}}$ to vary along with the linear model during the task-specific supervised model-tuning.

## 3.2 Data augmentations

We focus on random augmentations to protein primary sequences, both chemically and non-chemically motivated (see Fig. 1). Each of these data augmentations has reasons why it both might help generalize the model and might destroy the information contained in the primary sequence.

**Replacement (Dictionary/Alanine) [*RD & RA*].—** We randomly replace, with probability $p$, the $i^{\mathrm{th}}$ amino acid in the primary sequence $S = \{A_i\}_{i=1}^N$ with the most similar amino acid to it $A_i'$, according to a replacement rule; we do this independently for all $i$. We treat $p$ as a hyperparameter and will assess how $p$ affects downstream TAPE predictions. For Replacement (Dictionary), following French & Robson (1983), we pair each naturally-occuring amino acid with a partner that belongs to the same class (aliphatic, hydroxyl, cyclic, aromatic, basic, or acidic), but do not substitute anything for proline (only backbone cyclic), glycine (only an H side-chain), tryptophan (indole side chain), or histidine (basic side chain with no size or chemical reaction equivalent). We experimented with different pairings, finding little difference in the results; our best results were obtained with the final mappings: [[A,V], [S,T], [F,Y], [K,R], [C,M], [D,E], [N,Q], [V,I]]. We also study replacing residues with the single amino acid, alanine (A), as motivated by alanine-scanning mutagenesis (Cunningham & Wells, 1989). Single mutations to A are used experimentally to probe the importance of an amino acid because Alanine resembles a reduction of any amino acid to its $C_\beta$, which eliminates functionality of other amino acids while maintaining a certain backbone rigidity and is thus considered to be minimally disruptive to the overall fold of a protein, although many interesting exceptions can still occur by these mutations.

Table 1: Best linear evaluation results. **Bold** refers to cases that outperform the TAPE baselines; and red is the task-wise best-performing result. *MT* and *CL* refer to training with masked-token prediction and contrastive learning, respectively. Stability and fluorescence are scored by Spearman correlation and remote homology (fold, family, superfamily) and secondary structure (CASP12, TS115, CB513) by classification accuracy. Bootstrap errors are reported per task by taking the maximum error found for any of the models. (Also see Appendix C.)

| Scenario | Stability | Fluor. | Remote Homology | $2^{nd}$ Structure |
|---|---|---|---|---|
| MT: TAPE Baseline | 0.498 | 0.256 | [0.200, 0.625, 0.231] | [0.699, 0.756, 0.727] |
| MT: No Aug. ($\gamma = 0$) | **0.534** | **0.275** | [**0.206**, **0.636**, **0.241**] | [**0.706**, **0.771**, **0.729**] |
| MT: Best Aug. | **0.516** | **0.301** | [**0.207**, **0.637**, **0.241**] | [**0.716**, **0.771**, **0.735**] |
| CL: No Aug. | **0.512** | **0.334** | [0.146, 0.529, 0.163] | [0.667, 0.725, 0.678] |
| CL: Single Aug. | 0.562 | **0.343** | [0.183, **0.720**, **0.243**] | [0.700, 0.757, 0.727] |
| CL: Leave-one-out | 0.337 | **0.323** | [0.168, **0.686**, 0.222] | [**0.700**, 0.756, 0.723] |
| CL: Pairwise | **0.537** | **0.361** | [**0.219**, **0.718**, **0.255**] | [**0.702**, **0.759**, 0.726] |
| Bootstrap $1\sigma$ (<) | $\pm 0.011$ | $\pm 0.006$ | $\pm$[0.015, 0.014, 0.012] | $\pm$[0.021, 0.009, 0.015] |

**Global/Local Random Shuffling [*GRS & LRS*].—** We reshuffle the protein sequence, both globally and locally. For $S = \{A_i\}_{i=1}^N$, we define an index range $i \in [\alpha, \beta]$ with $\alpha < \beta \le N$, then replace amino acids $A_i$ in this range with a permutation chosen uniformly at random. We define Global Random Shuffling (GRS) with $\alpha = 1$ and $\beta = N$ and Local Random Shuffling (LRS) with the starting point of intervals chosen randomly between $\alpha \in [1, N-2]$ and $\beta = \min(N, \alpha + 50)$, ensuring at least two amino acids get shuffled. While shuffling aggressively destroys protein information, models trained with shuffling can focus more on permutation-invariant features, such as the overall amino acid counts and sequence length of the original protein.

**Sequence Reversion & Subsampling [*SR & SS*].—** For Sequence Reversion, we simply reverse the sequence: given $S = \{A_i\}_{i=1}^N$ we map $i \to i' = N - i$. Note that a protein sequence is oriented, proceeding sequentially from the N- to the C-terminii; reversing a protein sequence changes the entire structure and function of the protein. However, including reversed sequences might encourage the model to use short-range features more efficiently, as it has for seq2seq LSTM models (Sutskever et al., 2014). For Subsampling, we let the sequence index range from $i \in [\alpha, \beta]$, then uniformly sample $\alpha \in [1, N-2]$ and then preserve $A_i$, for $i = (\alpha, \alpha + 1, ..., \min(N, \alpha + 50))$. While many properties pertaining to the global fold of a protein are due to long-range interactions between residues that are well-separated in the primary sequence, properties such as proteosomal cleavage or docking depend more heavily on the local sequence regime, implying that training while prioritizing local features might still improve performance.

**Combining Augmentations.—** We consider applying augmentations to the fine-tuning of semi-supervised models both individually and together. For *Single Augmentations* we consider only one of the augmentations at a time. With *Leave-One-Out Augmentation* we define lists of augmentations to compare together; for each list of augmentations, we iteratively remove one augmentation from the list and apply all other augmentations during training. Finally, in *Pairwise Augmentation* all pairs of augmentations are considered.

## 4 RESULTS

**Assessing data-augmented representations with linear evaluation.—** The core result of this paper uses linear evaluation methods to probe how well data augmentations are able to improve the learned representation of protein sequences. In Table 1 we summarize the best results from various data augmentation procedures, highlighting all the cases that outperform the TAPE Baseline. We compare identical model architectures trained with the same data, but using various data augmen-

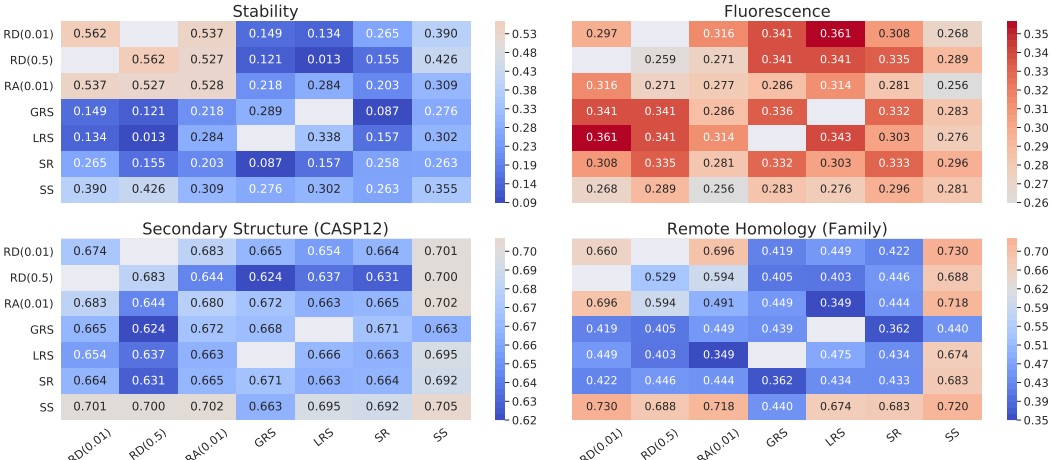

Figure 3: Contrastive learning performance with pairwise & single augmentations in linear evaluation for 4 different tasks. The axes refer to different augmentations, with diagonal being a single augmentation. The values in the heatmaps are correlation (stability and fluorescence) and classification accuracy (remote homology and secondary structure). We do not consider two pairs: RD($p = 0.01$) & RD($p = 0.5$) and GRS & LRS, due to redundancy. The per-task performance of the masked-token TAPE Baseline model is colored white in each subfigure; red is better performance, blue is worse. (Also see Appendix B.)

tations, to two baselines: (1) the transformer-based self-supervised model from Rao et al. (2019), which we call the TAPE Baseline; and (2) a contrastive learning model trained in the SimCLR approach, but using no data augmentations, *i.e.*, we only use the negative sampling part of SimCLR; we call this the Contrastive Baseline. We also report standard deviation ($\sigma$) for the major figures of this paper by bootstrapping the testing results 5,000 times, with convergence after $\sim 3,000$ samples.

We see broad improvement when using contrastive learning with data augmentations in comparison to both baselines for the stability, fluorescence, and remote homology tasks, and better-or-similar results for secondary structure prediction. For stability, we find *+0.064* in Spearman correlation ($\rho$) (best augmentation: RD($p = 0.01/0.5$)) in comparison to the TAPE Baseline and *+0.050* compared to the Contrastive Baseline. For fluorescence and remote homology, we see improvement to both the TAPE and Contrastive Baselines, with the pairwise combination of RD($p = 0.01$) & LSS) and RA($p = 0.01$) & SS yielding the best results, respectively. Compared to the TAPE Baseline, we obtain *+0.105* in $\rho$ for fluorescence and *[+1.9%, +9.3%, +2.4%]* in classification accuracy for remote homology on the three test sets; compared to the Contrastive Baseline, we find *+0.027* in $\rho$ and *[+7.3%, +18.9%, +9.2%]* in classification accuracy. The masked-token prediction model trained with RA($p = 0.01$) & SS performs the best on secondary structure, with *[+1.7%, +1.5%, +0.8%]* in the classification accuracy on the 3 test sets compared to the TAPE Baseline and *[+4.9%, +4.6%, +5.7%]* compared to the Contrastive Baseline. The best-performing contrastive learning model for secondary structure provides *[+3.5%, +3.4%, +4.8%]* in classification accuracy compared to Contrastive Baseline. Our interpretation is that augmentation and contrastive learning provide better encoded feature spaces that help improving the performance on protein's downstream tasks. See Appendix B for the complete results of linear evaluation experiments.

Figure 3 demonstrates our linear evaluation results using contrastive learning based on the composition of pairs of data augmentations. For stability, amino acid replacement (with either a dictionary (RD) or alanine alone (RA)) consistently improves performance compared to the TAPE baseline, as well as to other augmentation strategies, which typically underperform the baseline. Fluorescence sees improvements using all data augmentations, but random shuffling (LRS & GRS) as well as the binomial replacement of both types result in the best individual performance. For remote homology, it is apparent that subsampling plays an important role in model performance given the improvement it introduces on the three testing sets; the "family" homology level is included here and the other remote homology tasks are qualitatively similar. Similarly, we see that data augmentation

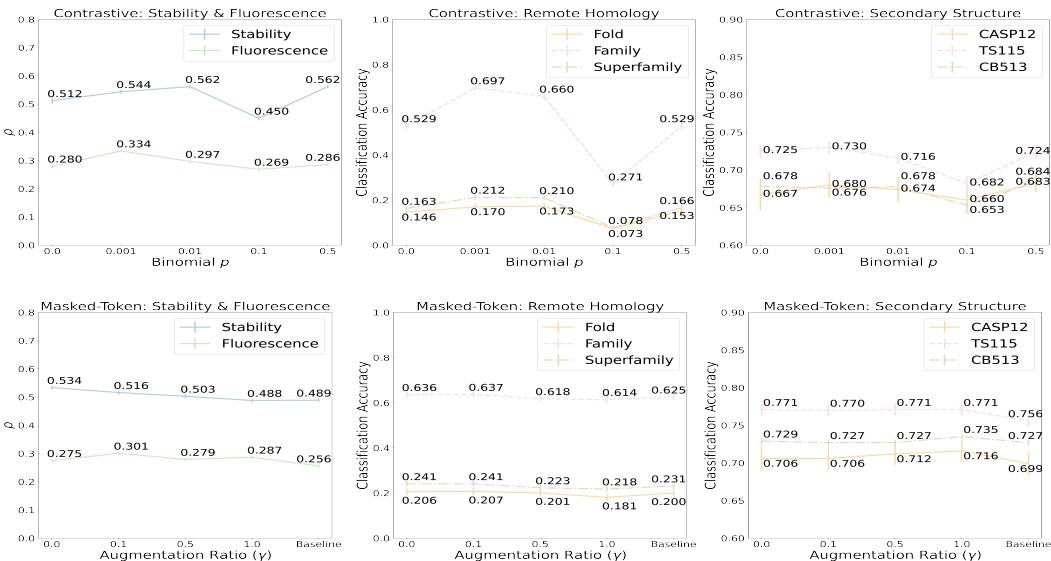

Figure 4: **Top row:** Effects of Binomial Replacement $p$ for linear evaluation on constrastive learning models for TAPE tasks. **Bottom row:** Effects of augmentation ratio $\gamma$ for linear evaluation on TAPE's self-supervised model with the best performing task-specific augmentations (see "Best Aug." row in Table 2) , using masked token prediction. The subfigures include linear evaluation results with different augmentation ratios, $\gamma$. "$\gamma$=0.0" refers to fine-tuning with no data augmentations, whereas "Baseline" refers to the TAPE pre-trained model with no further training. (Also see Appendix C.)

procedures that use subsampling tends to yield better performance than alternatives, with the best performing approach using subsampling alone. For complete heatmaps for the 6 remote homology and secondary structure testing sets, please refer to Fig. 5 in Appendix B.

**Effect of increasing augmentation rates.—** Fig. 4 presents results on the effect of varying data augmentations in two cases: (1) increasing the amino acid replacement probability $p$ for the Replacement Dictionary [*RD*] strategy with contrastive learning (top row); and (2) by augmenting increasingly larger fractions of the input data according to the best augmentations (see "Best Aug." row in Table 2) found in linear evaluation for masked-token prediction (bottom row). We define the augmentation ratio $\gamma$ as the fraction of the samples in the validation dataset that are randomly augmented per epoch; for contrastive learning fine-tuning, data augmentations are required for every data element.

For masked-token prediction, we see little change in performance for any task as a function of $\gamma$ using the for any of the best corresponding augmentation strategies. However, there is a small, but consistent, reduction in performance with increasing $\gamma$, implying that masked-token prediction is not always able to significantly improve its performance by using data augmentations. It is clear that large augmentation ratios $\gamma \sim 1$ hurt the model performance on Stability, Fluorescence and Remote Homology tasks. We also see that the TAPE Baseline model generally performs worse than further training with no data augmentation, indicating further training of the baseline model with the same data and procedure can improve the performance of Rao et al. (2019).

However, for contrastive learning, we see clear evidence that data augmentations can help generalization. We see increasing Spearman correlation for stability and fluorescence tasks and increasing accuracy for remote homology and secondary structure with increasing $p$ for $p < 0.01$. We see a consistent decrease in all metrics, to the lowest seen amount for each task, for replacement probability $p = 0.1$ and then a recovery to larger (sometimes the largest seen) for higher replacement $p = 0.5$. However, no augmentation, $p = 0$, consistently underperforms as compared to alternative values of $p > 0$.

Table 2: Model fine-tuning results, with associated training method and data augmentation procedure for each task. Testing sets for remote homology: (fold, family, superfamily), and for secondary structure: (CASP12, TS115, CB513).

| Scenario | Stability | Fluor. | Remote Homology | 2$^{nd}$ Structure |
|---|---|---|---|---|
| TAPE Best | 0.730 | 0.680 | [0.210, 0.880, 0.340] | [0.710, 0.770, 0.730] |
| Our Best | **0.748±0.005** | 0.677±0.004 | [0.209±0.015, **0.921±0.008, 0.377±0.014**] | [**0.711±0.015, 0.778±0.008, 0.739±0.003**] |
| Best Models | CL | CL | CL | MT |
| Best Aug. | RD(0.01 or 0.5) | RD(0.01) & LRS | RA(0.01) & SS | RA(0.01) & SS |

**Effect of contrastive learning.—** To assess the relative effects of contrastive learning and masked-token prediction, we compare results between the two approaches with or without data augmentations. All information for this comparison is in Fig. 4 and Table 1. It is unsurprising that using the identity function as a data transform in SimCLR (Eq. 1) yields little increase in generalizability; indeed, we see that masked-token prediction has better performance than contrastive learning for all tasks with no data augmentations (Fig. 4, $\gamma = 0$ vs $p = 0$). However, we see mixed results when comparing contrastive learning vs masked-token prediction methods with the same data augmentation techniques. As seen in Table 1 ("MR: Best Aug." row vs highest/red numbers in "CL: *" rows): contrastive learning significantly improves over masked-token prediction for stability (*+0.046* in $\rho$), fluorescence (*+0.061* in $\rho$), and remote homology (*[+1.2%, +8.1%, +1.4%]* in classification accuracy); and masked-token prediction improves over contrastive learning for secondary structure (*[+1.4%, +1.2%, +0.9%]* in classification accuracy). Overall, we cannot conclude from these pairs of linear evaluation studies that contrastive learning definitively performs better or worse than masked-token prediction on all downstream tasks: different tasks benefit from different training procedures and different combinations of data augmentations. However, we observe that the overall best results from Table 1 utilize the combination of contrastive learning with pairs of data augmentation for all tasks besides secondary structure prediction.

**Exploring the best performance via full fine-tuning.—** We provide results of the best performing fine-tuned models (on downstream tasks) and the comparison to the TAPE's original baselines in Table 2, in order to verify whether the learned representations of the best models provide good initialization points for transfer learning. Here, we have done full fine-tuning only on the best performing, per-task models found during the linear evaluation study (see Table 1). Notice that the baseline comparison changes in this table than for the linear evaluation results above because we allow the optimization to also adjust the parameters of the self-supervised models for every task (the TAPE baselines in Table 2 are from Rao et al. (2019)). The fine-tuned, data-augmented models outperform the TAPE baseline on stability (*+0.018* in Spearman correlation $\rho$), remote homology and secondary structure; and they perform within one $\sigma$ on fluorescence, although the large difference in performance between full fine-tuning and linear evaluation on the fluorescence tasks indicates that most of the model's predictive capacity is coming from the supervised learning task itself. The random amino acid replacement strategy is a consistent approach that achieved our best performance for all tasks and subsampling performed well on tasks that depend on the structural properties of proteins (remote homology and secondary structure): *[+0.0%, +4.1%, +3.7%]* classification accuracy for remote homology and *[+0.1%, +0.8%, +0.9%]* classification accuracy for secondary structure.

## 5 CONCLUSION

We experimentally verify that relatively naive string manipulations can be used as data augmentations to improve the performance of self-supervised protein sequence on the TAPE validation tasks. We demonstrate that, in general, augmentations will boost the model performance, in both linear evaluation and model fine-tuning cases. However, different downstream tasks benefit from different protein augmentations; no single augmentation that we studied was consistently the best. However, the approach that we have taken, where we fine-tune a pretrained model on the validation set re-

quires significantly lower computational cost than training on the full training set. Consequently, a modeler interested in a small number of downstream tasks would not be over-burdened to attempt fine-tuning of the semi-supervised model on a broad range of data augmentation transformations.

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

Table 3: Hyperparmeter Setups for Models in Different Tasks

| Levels | Hyperparmeters | | | | |
|---|---|---|---|---|---|
| | Batch Size | Learning Rate | # GPUs | # Epochs | Gradient Acc |
| SimCLR / TAPE Baseline | 512 | 1e-4/1e-5 | 8 | 30 | 8 |
| Stability | 256 | 1e-4/1e-5 | 1 | 60 | 4 |
| Fluorescence | 256 | 1e-4/1e-5 | 1 | 60 | 16 |
| Remote Homology | 256 | 5e-4/5e-5 | 1 | 60 | 8 |
| Secondary Structure | 256 | 1e-4/1e-5 | 1 | 40 | 8 |

## APPENDIX

## A   TRAINING DETAILS

For the augmented training, we focused on training the self-supervised part of the model. Namely, we apply the hyperparameters in Table 3, Row 1, on the self-supervised part in either the SimCLR (contrastive learning) or the masked-toke prediction model. Here we train all the models for 30 epochs on the Pfam validation set in order to make a relatively fair comparison. As discussed in the main paper, after the augmented training is finished, we perform linear evaluations on the pre-trained model in the previous steps with the hyperparameters listed in Table 3, Row 2-5 for the 4 downstream tasks. All of the linear evaluation results shown in the main paper and appendix are based on the best results we find after the augmented training and linear evaluation with the hyperparameters in Table 3. For model fine-tuning, since models trained with contrastive learning and with TAPE's semi-supervised learning models have different statistics (model parameters are different), we do not consider using the same set of hyperparameters. Instead, we report the best results in comparison to the TAPE's result. The corresponding hyperparameter setups of the best cases described above, including the augmented training and linear evaluation, can be found in Table 4. In addition, the optimizer being applied is AdamW, which is identical to the one in TAPE. Given the NVIDIA V100 GPUs we use have 16 GB memory, a memory limitation, we constrain the sequence length $\leq 512$ to enable the training. The batch size we report in appendix are the total batch that considers the number of GPUs. "Gradient Acc" in Table 3 is short for "Gradient Accumulation Steps", which describes how many steps per gradient update to the model.

## B   LINEAR EVALUATION RESULTS

Here we provide comprehensive linear evaluation results of contrastive learning with single augmentation, leave-one-out and pairwise augmentation. To Simplify the tables, we use the same abbreviations applied in the main paper to indicate the augmentations. We summarize the best results after the training and evaluation according to Section A for both single augmentation and pairwise augmentations in Figure 5. We use diverging palette with the center (gray/white) being the best linear evaluation baseline results with TAPE's pre-trained model and warmer colors refering to the better-than-baseline results and cooler colors being worse-than-baseline results. The diagonal values come from single augmentation setup and all other values come from pairwise augmentation setup. Table 5 and Table 6 also include Figure 5's corresponding values. By checking the values of the figure and tables, we observe the following: (1) For stability, the binomial replacement works well with single augmentation and pairwise augmentation. There is no improvement from leave-one-out augmentation cases. (2) For fluorescence, the pairwise augmentation with binomial replacement and shuffling can improve the model performance. (3) For remote homology, we clearly see improvements coming from subsampling on all of the three testing sets. And this is independent of other augmentations in the pairwise case. (4) For secondary structure, we do not observe gains from either single or pairwise augmentation, which is consistent with the discussion in the main paper. The best case for secondary structure comes from the masked-token prediction model with augmentations. Besides the results above, we also provide leave-one-out results in Table 8 with different leave-one-out cases that contain different augmentations. With leave-one-out augmentations, we observe only several cases where the leave-one-out cases outperform TAPE's baselines across 4 downstream

Table 4: Configurations of the best models

**Configurations of Best Models in Augmented Training**

|  | Stability | Fluorescence | Remote Homology | Secondary Structure |
|---|---|---|---|---|
| Model Type | SimCLR-based | SimCLR-based | SimCLR-based | TAPE-based |
| Batch Size | 512 | 512 | 512 | 512 |
| Gradient Acc | 8 | 8 | 8 | 8 |
| Learning Rate | 1e-5 | 1e-5 | 1e-5 | 1e-5 |
| # GPUs | 8 | 8 | 8 | 8 |
| # Epochs | 30 | 30 | 30 | 30 |
| Aug. Type | Single | Pairwise | Pairwise | Pairwise |
| Aug. Config | $RD(p = 0.01/0.50)$ | $RD(p = 0.01)$ + LRS | RA + SS | RA + SS |

**Configurations of Best Models in Linear Evaluation**

|  | Stability | Fluorecsence | Remote Homology | Secondary Structure |
|---|---|---|---|---|
| Model Type | SimCLR-based | SimCLR-based | SimCLR-based | TAPE-based |
| Batch Size | 256 | 256 | 256 | 64 |
| Gradient Acc | 4 | 4 | 16 | 8 |
| Learning Rate | 1e-4 | 1e-4 | 5e-4 | 1e-4 |
| # GPUs | 1 | 1 | 1 | 1 |
| # Epochs | 60 | 60 | 60 | 40 |

**Configurations of Best Models in Fine-tuning**

|  | Stability | Fluorecsence | Remote Homology | Secondary Structure |
|---|---|---|---|---|
| Model Type | SimCLR-based | SimCLR-based | SimCLR-based | TAPE-based |
| Batch Size | 256 | 32 | 32 | 64 |
| Gradient Acc | 4 |  | 16 | 16 |
| Learning Rate | 1e-4 | 3e-5 | 3e-5 | 1e-4 |
| # GPUs | 1 | 1 | 1 | 1 |
| # Epochs | 60 | 60 | 20 | 50 |

Table 5: Linear Evaluation Results of Single Augmentation

| Scenario | Stability | Fluorescence | Remote Homology | Secondary Structure |
|---|---|---|---|---|
| Baseline | 0.489 | 0.256 | [0.200, 0.625, 0.231] | [0.699, 0.756, 0.727] |
| $RD(p = 0.01)$ | **0.562** | **0.297** | [0.173, **0.660**, 0.210] | [0.674, 0.713, 0.679] |
| $RD(p = 0.5)$ | **0.562** | 0.259 | [0.153, 0.529, 0.166] | [0.683, 0.724, 0.684] |
| $RA(p = 0.01)$ | **0.528** | **0.277** | [0.164, 0.491, 0.160] | [0.680, 0.706, 0.674] |
| GRS | 0.289 | **0.336** | [0.110, 0.439, 0.100] | [0.668, 0.683, 0.646] |
| LRS | 0.338 | **0.343** | [0.100, 0.475, 0.095] | [0.666, 0.679, 0.646] |
| SR | 0.258 | **0.333** | [0.104, 0.433, 0.200] | [0.664, 0.682, 0.646] |
| SS | 0.355 | **0.281** | [0.183, **0.720**, **0.243**] | [**0.705**, 0.735, 0.707] |

tasks. Nevertheless, the leave-one-out results provide insights that one should not combine arbitrary augmentations given the decrease in the performance we observe in leave-one-out cases.

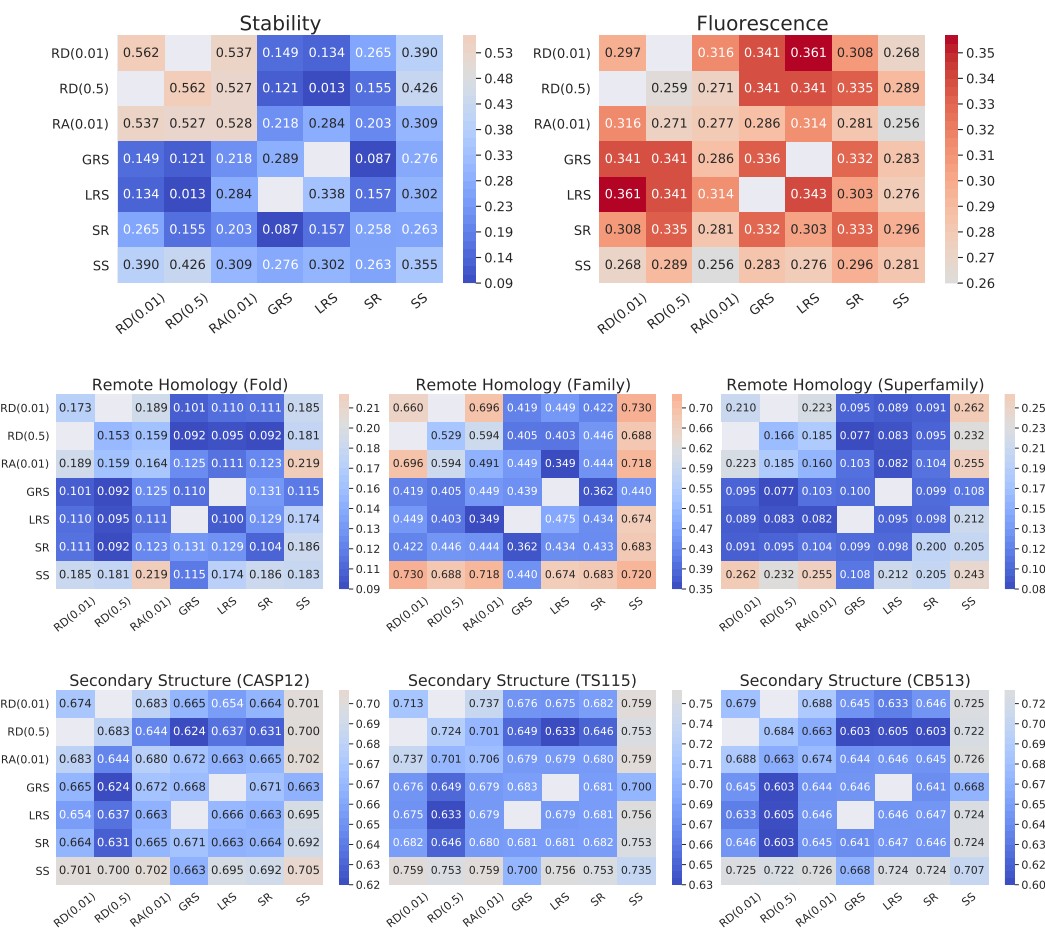

Figure 5: Contrastive Learning Performance with Pairwise & Single Augmentation in Linear Evaluation. The heatmaps include the performance of contrastive learning models with pairwise/single augmentations for 4 different downstream tasks considering all possible testing sets. Both x and y axes refer to different augmentations. The diagonal of heatmaps refer to the single augmentation cases. All other cells refer to cases with pairwise augmentations. The values in the heatmaps refer to evaluation results: "$\rho$" for Stability and Fluorescence, and "Classification Accuracy" for Secondary Structure and Remote Homology.

Table 6: Linear Evaluation Results of Pairwise Augmentation

| Scenario | Stability | Fluorescence | Remote Homology | Secondary Structure |
|---|---|---|---|---|
| Baseline | 0.489 | 0.256 | [0.200, 0.625, 0.231] | [0.699, 0.756, 0.727] |
| RD($p = 0.01$) & RA($p = 0.01$) | **0.537** | **0.316** | [0.189, **0.696**, 0.223] | [0.683, 0.737, 0.688] |
| RD($p = 0.01$) & GRS | 0.149 | **0.341** | [0.101, 0.419, 0.095] | [0.665, 0.676, 0.645] |
| RD($p = 0.01$) & SR | 0.265 | **0.308** | [0.111, 0.422, 0.091] | [0.664, 0.682, 0.646] |
| RD($p = 0.01$) & LRS | 0.134 | **0.361** | [0.097, 0.449, 0.089] | [0.654, 0.675, 0.633] |
| RD($p = 0.01$) & SS | 0.390 | **0.268** | [0.185, **0.730**, **0.262**] | [**0.701**, **0.759**, 0.725] |
| RD($p = 0.5$) & RA($p = 0.01$) | **0.527** | **0.271** | [0.159, 0.594, 0.185] | [0.644, 0.701, 0.663] |
| RD($p = 0.5$) & GRS | 0.121 | **0.341** | [0.092, 0.405, 0.077] | [0.624, 0.649, 0.603] |
| RD($p = 0.5$) & SR | 0.155 | **0.335** | [0.092, 0.446, 0.095] | [0.631, 0.646, 0.603] |
| RD($p = 0.5$) & LRS | 0.013 | **0.341** | [0.095, 0.403, 0.083] | [0.637, 0.633, 0.605] |
| RD($p = 0.5$) & SS | 0.426 | **0.289** | [0.181, **0.688**, 0.232] | [**0.700**, **0.753**, 0.722] |
| RA($p = 0.01$) & GRS | 0.218 | **0.286** | [0.125, 0.449, 0.103] | [0.672, 0.679, 0.644] |
| RA($p = 0.01$) & SR | 0.203 | **0.281** | [0.123, 0.444, 0.104] | [0.665, 0.680, 0.645] |
| RA($p = 0.01$) & LRS | 0.284 | **0.314** | [0.111, 0.349, 0.082] | [0.663, 0.679, 0.646] |
| RA($p = 0.01$) & SS | 0.309 | 0.256 | [**0.219**, **0.718**, **0.255**] | [**0.702**, **0.759**, 0.726] |
| GRS & SR | 0.087 | **0.332** | [0.131, 0.362, 0.099] | [0.671, 0.681, 0.641] |
| GRS & SS | 0.276 | **0.283** | [0.115, 0.440, 0.108] | [0.663, 0.700, 0.668] |
| SR & LRS | 0.157 | **0.303** | [0.129, 0.434, 0.098] | [0.663, 0.681, 0.647] |
| SR & SS | 0.263 | **0.296** | [0.186, **0.683**, 0.205] | [0.692, **0.753**, 0.724] |
| LRS & SS | 0.302 | **0.276** | [0.174, **0.674**, 0.212] | [0.695, **0.756**, 0.724] |

Table 7: Best linear evaluation results. **Bold** refers to cases that outperform the TAPE baselines; and **red** is the task-wise best-performing result. *MT* and *CL* refer to training with masked-token prediction and contrastive learning, respectively. Stability and fluorescence are scored by Spearman correlation ($\rho$) and remote homology and secondary structure by *Cross Entropy*. Bootstrap errors are reported per task by taking the maximum error found for any of the models.

| Scenario | Stability | Fluor. | Remote Homology | $2^{nd}$ Structure |
|---|---|---|---|---|
| MT: TAPE Baseline | 0.498 | 0.256 | [4.433, 1.862, 4.378] | [0.661, 0.541, 0.616] |
| MT: No Aug. ($\gamma = 0$) | **0.534** | **0.275** | [4.445, **1.860**, 4.381] | [**0.643**, **0.532**, **0.608**] |
| MT: Best Aug. | **0.516** | **0.301** | [4.437, 1.872, 4.404] | [**0.649**, **0.529**, **0.602**] |
| | | | | |
| CL: No Aug. | **0.512** | **0.334** | [4.884, 2.283, 4.900] | [0.753, 0.625, 0.713] |
| CL: Single Aug. | **0.562** | **0.343** | [4.709, **1.285**, **4.214**] | [0.672, 0.561, 0.649] |
| CL: Leave-one-out | 0.337 | **0.323** | [5.001, **1.479**, 4.460] | [0.705, 0.574, 0.674] |
| CL: Pairwise | **0.537** | **0.361** | [4.433, **1.279**, **4.045**] | [0.676, 0.560, 0.638] |
| | | | | |
| Bootstrap $1\sigma$ ($<$) | $\pm 0.011$ | $\pm 0.006$ | $\pm$[0.010, 0.005, 0.007] | $\pm$[0.038, 0.018, 0.009] |

## C  SUPPLEMENTARY

In this section, we provide supplementary results. Specifically, we provide the complete comparison table on the masked-token prediction model with different augmentation ratio $\gamma$ in Table 9. We also provide the corresponding plot in Figure 4 and analylsis in the main paper. Besides, Table 7 provides similar information as Table 2 in the main paper, except for values for remote homolody and secondary structure here being the cross entropy, rather than classification error.

Table 8: Linear Evaluation Results with Leave-one-out Augmentations

**Meta Description**

| Cases | Leave-one-out Augmentations |
|---|---|
| Case 1 | [RD($p = 0.01$), RA, GRS, SR, SS] |
| Case 2 | [RD($p = 0.50$), RA, GRS, SR, SS] |
| Case 3 | [RD($p = 0.01$), RA, LRS, SR, SS] |
| Case 4 | [RD($p = 0.50$), RA, LRS, SR, SS] |

**Stability**

| | Case 1 | Case 2 | Case 3 | Case 4 |
|---|---|---|---|---|
| All entries | 0.337 | 0.161 | 0.212 | 0.248 |
| No RD(0.01 or 0.5) | 0.230 | 0.230 | 0.335 | 0.230 |
| No RA | 0.136 | 0.113 | 0.207 | 0.195 |
| No GRS or LRS | 0.222 | 0.088 | 0.222 | 0.088 |
| No SR | 0.326 | 0.145 | 0.243 | 0.138 |
| No SS | 0.103 | -0.082 | 0.054 | -0.018 |
| MT: TAPE Baseline | 0.489 | | | |

**Fluorescence**

| | Case 1 | Case 2 | Case 3 | Case 4 |
|---|---|---|---|---|
| All entries | **0.293** | **0.314** | **0.281** | **0.281** |
| No RD(0.01 or 0.5) | 0.250 | 0.250 | **0.284** | 0.250 |
| No RA | 0.229 | 0.222 | **0.280** | **0.283** |
| No GRS or LRS | **0.288** | **0.323** | **0.288** | **0.323** |
| No SR | 0.260 | **0.268** | **0.286** | **0.308** |
| No SS | **0.288** | **0.263** | **0.308** | 0.251 |
| MT: TAPE Baseline | 0.256 | | | |

**Remote Homology**

| | Case 1 | Case 2 | Case 3 | Case 4 |
|---|---|---|---|---|
| All entries | [0.104, 0.435, 0.102] | [0.091, 0.407, 0.097] | [0.132, 0.624, 0.179] | [0.128, 0.543, 0.153] |
| No RD(0.01 or 0.5) | [0.111, 0.441, 0.100] | [0.111, 0.441, 0.100] | [0.128, 0.614, 0.178] | [0.111, 0.441, 0.100] |
| No RA | [0.108, 0.433, 0.106] | [0.085, 0.411, 0.099] | [0.125, 0.615, 0.178] | [0.130, 0.537, 0.148] |
| No GRS or LRS | [0.166, **0.688**, 0.214] | [0.152, 0.609, 0.180] | [0.166, **0.688**, 0.214] | [0.152, 0.609, 0.180] |
| No SR | [0.105, 0.436, 0.100] | [0.084, 0.296, 0.086] | [0.168, **0.686**, 0.222] | [0.135, **0.636**, 0.192] |
| No SS | [0.104, 0.428, 0.08] | [0.105, 0.339, 0.086] | [0.105, 0.435, 0.089] | [0.095, 0.363, 0.084] |
| MT: TAPE Baseline | [0.200, 0.625, 0.231] | | | |

**Secondary Structure**

| | Case 1 | Case 2 | Case 3 | Case 4 |
|---|---|---|---|---|
| All entries | [0.664, 0.680, 0.648] | [0.671, 0.702, 0.673] | [0.690, 0.750, 0.718] | [0.691, 0.744, **0.715**] |
| No RD(0.01 or 0.5) | [0.659, 0.701, 0.666] | [0.659, 0.701, 0.666] | [0.688, 0.751, 0.720] | [0.659, 0.701, 0.666] |
| No RA | [0.663, 0.701, 0.667] | [0.581, 0.588, 0.567] | [0.125, 0.615, 0.178] | [0.685, 0.745, 0.715] |
| No GRS or LRS | [0.695, 0.754, 0.723] | [0.687, 0.746, 0.718] | [0.695, 0.754, 0.723] | [0.152, 0.609, 0.180] |
| No SR | [0.673, 0.703, 0.668] | [0.670, 0.705, 0.675] | [**0.700**, 0.756, 0.723] | [**0.700**, 0.747, 0.719] |
| No SS | [0.665, 0.684, 0.645] | [0.638, 0.658, 0.615] | [0.665, 0.682, 0.645] | [0.630, 0.658, 0.624] |
| MT: TAPE Baseline | [0.699, 0.756, 0.727] | | | |

Table 9: Ratio Effects on the Masked-Token Model

| Scenarios | $\gamma$ | Results | Baselines | Augmentations |
|---|---|---|---|---|
| Stability | 0.1 | **0.516±0.007** | | |
| | 0.5 | **0.503±0.007** | 0.489±0.007 | RD($p = 0.01/0.5$) |
| | 1.0 | 0.488±0.007 | | |
| Fluorescence | 0.1 | **0.301±0.005** | | |
| | 0.5 | **0.279±0.006** | 0.256±0.006 | RD($p = 0.01$) & LRS |
| | 1.0 | **0.287±0.006** | | |
| Remote Homology | 0.1 | [**0.207±0.015**, **0.637±0.014**, **0.241±0.012**] | | |
| | 0.5 | [**0.201±0.015**, 0.618±0.013, 0.223±0.012] | [0.200±0.014, 0.625±0.014, 0.231±0.012] | RA($p = 0.01$) & SS |
| | 1.0 | [0.181±0.015, 0.614±0.015, 0.218±0.012] | | |
| Secondary Structure | 0.1 | [**0.706**±0.016, **0.770±0.008**, 0.727±0.004] | | |
| | 0.5 | [**0.712±0.016**, **0.771±0.008**, 0.727±0.004] | [0.699±0.016, 0.756±0.008, 0.727±0.004] | RA($p = 0.01$) & SS |
| | 1.0 | [**0.716±0.015**, **0.771±0.008**, **0.735±0.003**] | | |

