# OpenReview forum: "Improving Generalizability of Protein Sequence Models via Data Augmentations"
_ICLR.cc/2021/Conference — Reject_

### Official Review · AnonReviewer2 · 2020-10-27
**A paper tackling an important topic in protein deep learning**

**Rating:** 6
**Confidence:** 3

**Review:**

### Summary

- This paper experimentally investigates the effects of data augmentation for protein deep learning.
- The authors first pretrained a transformer-based NN without data augmentation; then trained the model using data augmentation; finally, each TAPE task was fine-tuned.
- For data augmentation strategies, they used the Random dictionary, Random Alanine, Global/Local Shuffling, Sequence Reversion, and Subsampling.
- In the data augmentation phase, the Masked-token prediction (MT) and the Contrastive Learning (CL) have been studied.

### Strong Points

- I agree that investigating the data augmentation effect is important in protein deep learning because effective strategies are not very straightforward compared to other application domains, such as image processing.
- Experiments are conducted against wide downstream tasks. The authors found that CL with random replacement would be an effective data augmentation.
- The paper is well-written and easy to follow.

### Weak Points

- Randomness in the experiments: Are the reported values averaged over several attempts with different random seeds (i.e., initial weight of neural network, or randomness in data augmentation)? The authors have (seemingly) reported standard deviation only by the bootstrap over test data. I think this is important because, for example in Fig.4, the reported values seem to be affected by randomness (i.e., at p=0.1); the authors do not provide a reason for this (so I understood that it comes from randomness).
- Possibility of the other data augmentation: This paper argues that the random replacement strategies (RD/RA) work well for downstream tasks. However, this direction is not investigated deeply. It would be better if the authors will compare it with other replacement strategies. For example:
  - What happens if we use the other amino acid mappings described in Section 3.2? The authors reports only the best mapping, i.e., [[A,V], [S,T], [F,Y], [K,R], [C,M], [D,E], [N,Q], [V,I]]; how is the impact when using the other mapping?
  - What happens if we replace an amino acid with the other one that is uniform-randomly chosen from the other 19 possibilities? (Although the authors do not investigate it due to the lack of biological reasons, the following paper reports that this strategy also improves the accuracy for the secondary structure prediction task; Neural Edit Operations for Biological Sequences, NeurIPS 2018)
- The insights found and argued in this paper are not very strong. Actually, the authors say "different downstream tasks benefit from different protein augmentations" at the conclusion. This is a very well-known fact in the deep-learning community.


### Minor Points

- In "Replacement (Dictionary/Alanine)" paragraph in Section 3.2, the period "(A)." should be removed.


### Evaluation

Overall, this paper provides an interesting direction for protein learning, where effective data augmentation strategies are not very straightforward compared to other application domains (e.g., image processing). Although I agree that this is an important topic, I have several concerns as mentioned in the "Weak Points" above. Please clarify and address my concerns.

---

> ### Author Response · Authors · 2020-11-21
> **RE: Reviewer2**
>
> Thanks for the recognition of this work! Please refer to the comments below:
>
> **1. Randomness in the experiments:** *Are the reported values averaged over several attempts with different random seeds (i.e., initial weight of neural network, or randomness in data augmentation)? The authors have (seemingly) reported standard deviation only by the bootstrap over test data. I think this is important because, for example in Fig.4, the reported values seem to be affected by randomness (i.e., at p=0.1); the authors do not provide a reason for this (so I understood that it comes from randomness)*
>
> **ANSWER:** Yes, the standard deviations reported in the results are calculated based on bootstrapping the testing set for the corresponding identified models presented in the paper. To explore the effects of randomness in the hyperparameter setup and random initialization, we explore 4 different hyperparameter setups for each of the configurations presented in the paper (see Table 3.). And the results, e.g. shown in figure 4, are the bootstrapping results from the best models in the 4 experiments in each of the augmentation configurations. We agree that marginalizing over different initial conditions for the training would increase the errors bars over what we report here, but disagree that this is the correct approach, given the known sensitivity of optimizing well-performing attention models to their random initialization.  We believe that restricting to a small number of training runs (4) protects sufficiently against overfitting and underfitting.
>
> **2. Random replacement strategies:**
>
> - *What happens if we use the other amino acid mappings described in Section 3.2? The authors reports only the best mapping, i.e., [[A,V], [S,T], [F,Y], [K,R], [C,M], [D,E], [N,Q], [V,I]]; how is the impact when using the other mapping?*
>
>     **ANSWER:** We explored different mappings motivated by conservative substitution, but observed relatively equivalent performance across mappings we explored. The differences are small and we did not pursue them in depth for the detailed studies in Fig 4, for example.  Because of this reason, we choose to present only one mapping to show, in general, the performance of this type of augmentation.
>
> - *What happens if we replace an amino acid with the other one that is uniform-randomly chosen from the other 19 possibilities? (Although the authors do not investigate it due to the lack of biological reasons, the following paper reports that this strategy also improves the accuracy for the secondary structure prediction task; Neural Edit Operations for Biological Sequences, NeurIPS 2018)*
>
>     **ANSWER:** We agree this could be explored, too, as could many other alternative string manipulations.  It was our feeling that this would not be sufficiently distinct from “Replacement Dictionary” and “Replacement Alanine” to warrant inclusion.
>
> **3. Insights:**
> *The insights found and argued in this paper are not very strong. Actually, the authors say "different downstream tasks benefit from different protein augmentations" at the conclusion. This is a very well-known fact in the deep-learning community.*
>
> **ANSWER:** We disagree that the insights we derive from this study are not strong, although we understand that it would be simpler if we were to have found that one type of transformation were universally best across all tasks.  One of the main messages we deliver in our paper is that we have found that 2 types of augmentations (biologically-motivated replacement & local sub-sampling) perform well on related downstream tasks that fall into the “non-structural” and “structural” classes, respectively.  This is an important result, as most protein modeling use-cases can be roughly divided into those instances where one has structural information available or is attempting to predict structural properties and where one does not.  This is fundamentally different than finding that each related task could be improved by unrelated transforms.

---

### Official Review · AnonReviewer3 · 2020-10-27
**Missing key components**

**Rating:** 4
**Confidence:** 4

**Review:**

The authors suggest that training self-supervised protein sequence models with string manipulations as data augmentations will lead to better downstream performance.
They study several different types of data augmentations, (1) dictionary replacement, (2) global/local random shuffling, (3) sequence reversion and subsampling, and (4) a combination thereof.
The authors show promising results, where contrastive training improves benchmark results on 3 of the four tasks proposed in TAPE.
Additionally, data augmentations show better results on all tasks, beating the TAPE baseline significantly.

I find it exceptionally interesting that sequence reversal has such a negative impact on pretraining.
I would never have expected the order of protein sequence has such a large effect on downstream tasks.
Additionally, this is a very good line of work to pursue, since contrastive methods are known to work on images, but have not shown to beat MLM on NLP tasks.
It is interesting to find out whether they do well on protein language modeling tasks.

However, there are a few glaring weaknesses in the paper.

(1)
The authors do not even attempt to test on contact prediction, citing a data embargo.
In fact, many of the CASP12 structures are available, and are tested on in various papers, including the TAPE baseline, trRosetta, etc.
I believe contact prediction is a main problem to show gains on out of the 5 TAPE tasks, and without showing it in some form, this paper should not be accepted. (See CASP competitions)

(2)
The results are reported to be best across all data augmentations taking the best per task.
This seems unfair, as the representations learned should not depend on the downstream tasks.
Since each augmentation seem to have extremely different effects on each task, I would not say that on the whole data augmentation is actually useful in building a protein representation.
For example, checking the best model for stability, RD(0.01), I find that this augmentation only does better than the baseline on fluorescence and RH@family.

(3)
I have a few worries about the augmentation strategies used.
- For dictionary replacement, a much more natural choice is to follow probabilities outlined in BLOSUM or PAM substitution matrices.
  I believe this would be a stronger result than Alanine substitution, though the hand designed replacement scheme is quite interesting.
- GRS and LRS do not make much sense to me.
  It's quite unnatural to shuffle amino acids, and it's surprising to me that these representations would do as well as it does on secondary structure.
  In fact, I don't think GRS should be able to do better than a linear model on just amino acid identities on secondary structure prediction, so either there is a bug here or this is a quirk of the dataset.
  I also think LRS with a percentage of the sequence length is more interesting, and probably ends up being similar to the MLM loss if you use 15%.
- In figure 4 - it's not clear why the contrastive models all have a drop in performance at p=0.1. I don't see any reason that this should happen across multiple tasks and augmentation strategies.


I also offer some small suggestions for a more clear and better paper:
- Sec 2, last sentence of paragraph 1 claims protein sequences do not need to preserve contextual meaning.
  This is simply wrong.
  They must preserve the ability to fold into useful biological structures, and augmentation strategies like random shuffling remove this property - please remove this sentence.
- The 'MT' task is more commonly referred to in literature as MLM (Masked Language Modeling), I suggest to use this term instead.
- Figure 3's color scheme is counter-intuitive and not consistent across tasks.
  Why is gray the best on two tasks, where as light red is best on remote homology and dark red best on fluorescence?
  Please use a more intuitive color scheme, where perhaps the bold colors show where the best results are.
- Please permute the numbers for remote homology so it is ordered (fold, superfamily, family) - this is the natural ordering for this task.
- Table 7: why are the linear evaluation results now reported by cross entropy rather than classification accuracy as in previous tables?


As a summary, my main criticisms of this paper is (1) not including contact prediction, (2) picking the data augmentation to use for each task, and (3) augmentations with unreasonable intuitions.

Edit: Read responses, bumped score up to a 4, check full response below

---

> ### Author Response · Authors · 2020-11-21
> **RE: Reviewer3 (part2)**
>
> **3. Worries about Augmentation Strategies:**
> - *For dictionary replacement, a much more natural choice is to follow probabilities outlined in BLOSUM or PAM substitution matrices. I believe this would be a stronger result than Alanine substitution, though the hand designed replacement scheme is quite interesting.*
>
>     **ANSWER:** We intentionally did not pursue BLOSUM-based substitutions, as these rely on the multiple sequence alignment, which may or may not be of high quality for a given protein sequence.   Furthermore, it requires a non-trivial pre-processing step and has a dependence on the hyperparameters of the alignment algorithm (e.g., the BLAST e-value cutoffs).  We agree it would be interesting and should be pursued in the future.
>
> - *GRS and LRS do not make much sense to me. It's quite unnatural to shuffle amino acids, and it's surprising to me that these representations would do as well as it does on secondary structure. In fact, I don't think GRS should be able to do better than a linear model on just amino acid identities on secondary structure prediction, so either there is a bug here or this is a quirk of the dataset. I also think LRS with a percentage of the sequence length is more interesting, and probably ends up being similar to the MLM loss if you use 15%.*
>
>     **ANSWER:** Global random shuffling (GRS) and sequence reversion are used here as a quasi- null-model, as we agree that these are going to radically alter the protein sequence unnaturally.  We’ll modify the language of the paper to more clearly articulate why we included these.  However, we disagree about local random shuffling (LRS).  We feel that performing unnatural, information-destroying mutations in local contiguous regions of the protein sequence, require the semi-supervised models to focus on non-mutated regions of the protein during training.  If the important signal for a task is contained in local features, then this is a procedure to highlight their importance during training. We don’t exactly follow your suggestion about LRS at 15% being equivalent to MLM loss.
>
> - *In figure 4 - it's not clear why the contrastive models all have a drop in performance at p=0.1. I don't see any reason that this should happen across multiple tasks and augmentation strategies.*
>
>     **ANSWER:** The drop in performance at p=0.1 in Fig 4 is not that surprising to us.  In fact, it was our expectation that there would be a “sweet spot” where we have performed just enough amino acid substitutions to improve generalizability, but not so much that the original protein sequence is unrecognizable.  What is more surprising is the increasing performance at higher p=0.5.  We initially suspected randomness at the p=0.5 result, but the same relative ordering appeared in all 4 training runs with only trivial differences in the y-axis; we leave this here as an empirical artifact for the reader to interpret.  However, one important point to note is that this figure does not only focus on augmentations themselves, but also on the joint effects of augmentations and contrastive learning. And the drop in performance at p=0.1 and subsequent rise at p=0.5 is not seen in the MT (or MLM) model.
>
> - *Sec 2, last sentence of paragraph 1 claims protein sequences do not need to preserve contextual meaning. This is simply wrong. They must preserve the ability to fold into useful biological structures, and augmentation strategies like random shuffling remove this property - please remove this sentence. The 'MT' task is more commonly referred to in literature as MLM (Masked Language Modeling), I suggest to use this term instead.*
>
>     **ANSWER:** Thanks for pointing this out. This last sentence focuses on the differences between the augmentation for NLP and augmentation for protein. The “contextual preservation” is for NLP sentences, which is used for the comparison to protein.
>
> - *Figure 3's color scheme is counter-intuitive and not consistent across tasks...*
>
>     **ANSWER:** We use a divergent color scheme to try to indicate better/worse than the masked-token model. And the masked token model is marked in gray.
>
> We acknowledge your worries on the augmentation strategies and address your points below.  However, we want to make it clear that if the types of simple data augmentations we study in this paper are successful, it inevitably leads to follow-up questions that critically examine the specifics of the data augmentation procedures and encourages suggestions of alternate data augmentation methods.  For an empirical paper, we view this as a feature, not a bug. Scientific inquiry starts with the simplest hypotheses. If the simplest augmentations are successful, this clears a major hurdle for a new research avenue into alternative data augmentation procedures of varying complexity.

---

> ### Author Response · Authors · 2020-11-21
> **RE: Review3 (part1)**
>
> Thanks for the recognition and appreciation of this work! Here are the comments to the questions raised above:
>
> **1. CASP 12 and Contact Prediction:** *The authors do not even attempt to test on contact prediction, citing a data embargo. In fact, many of the CASP12 structures are available, and are tested on in various papers, including the TAPE baseline, trRosetta, etc. I believe contact prediction is a main problem to show gains on out of the 5 TAPE tasks, and without showing it in some form, this paper should not be accepted. (See CASP competitions)*
>
> **ANSWER:** As you indicate, the absence of this task is due to caveats around the contact prediction dataset given by the TAPE authors, which is prepared based on the ProteinNet repo (https://github.com/aqlaboratory/proteinnet) that still indicates substantial quantities of high-quality structures embargoed from this version of the testing set.  To avoid the unknown effects of the embargoed structures on the contact prediction assessment, we made a judgment call to remove it and perform no experiments on this task.  This was a hard decision for us, because we agree that we could have just included them in our study, likely with little change to our conclusions given the consistency of the local random sub-sampling augmentation on the other structural tasks.  However, as you pointed out by mentioning the CASP competitions, we did not feel that including this task would even provide a fair comparison to the latest round of high-performing CASP12 ML models, such as AlphaFold and RaptorX, which were evaluated on the non-embargoed version of these data.  We thought it cleanest to just omit it.
> Additionally, we point out this paper’s major focus is on data augmentation and contrastive learning; we feel that our experiments on the remaining 4 TAPE tasks are sufficient to convey our conclusions, especially given that we have shown results on 2 other tasks related to protein structure.
>
>
>
>
> **2. Best Augmentations:** *The results are reported to be best across all data augmentations taking the best per task. This seems unfair, as the representations learned should not depend on the downstream tasks. Since each augmentation seem to have extremely different effects on each task, I would not say that on the whole data augmentation is actually useful in building a protein representation. For example, checking the best model for stability, RD(0.01), I find that this augmentation only does better than the baseline on fluorescence and RH@family.*
>
> **ANSWER:** One of the main findings of our paper is that we have identified two augmentations (biologically-motivated replacement & local sub-sampling) that are clearly better performing on the non-structural and structural families of TAPE tasks, respectively.  We do agree that, given the data we have available, our study demonstrates that protein invariances are inherently more task dependent than we see in other ML fields such as CV or NLP.  However, in just these TAPE validation tasks we are asking semi-supervised protein models to learn representations that generalize to wildly diverse areas, including non-natural/engineered proteins, structural arrangement of proteins, and proteins that exist in extremely different environments (aqueous vs membrane, for example) - none of this is obvious from the protein sequence itself.  In all of these cases we have seen that a handful of easy-to-implement string manipulations can improve the representations, especially when we do not allow the representations to vary during task-specific fine-tuning (linear evaluation, Table 1).  In addition, the information destroying augmentations did provide improvements on some protein tasks (see. Table2.), which we find interesting and may point to pathologies in the way deep attentional models generalize to new protein tasks. This provides insights for people in the protein modeling area that when training models, it is also worthwhile to think of some augmentations to destroy some local protein features during the modeling. We believe these two findings are vital, even if they are not as simple as saying “Augmentation Strategy X is always the best”.

---

### Official Review · AnonReviewer1 · 2020-10-28
**Incremental; lack of rigor**

**Rating:** 3
**Confidence:** 4

**Review:**

A suite of data augmentations is presented for improving protein language models.

Strengths
- Data augmentation and contrastive models are underexplored areas in protein language models. The authors are able to improve upon the TAPE baseline. It would be interesting to see if these data augmentations could be used on recent models (ESM-1 and ProTrans) to achieve a new SOTA.

Weaknesses
- All evaluation was performed on the TAPE model, which was released >1 year ago. Since then, there have multiple models released that have much better performance (ProTrans from Elnaggar, et al.; ESM-1 from Rives, et al.).
- Related work is incomplete. For example, it says SimCLR is the current SOTA contrastive learning technique. There is already SimCLR-v2 that should be cited, among others. Even though the authors use the models from Rao, et al. 2019, the authors should also cite Alley, et al. 2019, Heinzinger et al. 2019, and Rives, et al. 2019 which were first to introduce semi-supervised learning for protein language models.
- The data augmentation is done on the validation set. This means there is no longer a validation set! Their model performs better than the ones presented by Rao, et al. but that could be because additional data was incorporated here. The authors include a baseline where they continued training Rao, et al. on the validation set, but it would be better practice to augment data from the training set.
- The authors do not evaluate on contact prediction because CASP12 "has an incomplete test set due to data embargoes." However, CASP12 ended 4 years ago and the data is now available. In fact, the authors use CASP12 for secondary structure prediction, so clearly they have access to the data. Furthermore, the authors could consider additional structural sets, such as structural holdout at family level; temporal holdout to CASP12/CASP13; other at the very least a sequence based split.
- The data shuffling will be predominantly shifted to the right because beta = min(N, alpha + 50).

Additional
- In the abstract, the authors refer to fine-tuning semi-supervised protein models. This is not very specific, as the authors actually continue the pre-training procedure with the data augmentations.
- For the replacement augmentation, why do you randomly replace with a single amino acid instead of rotating between, e.g. all hydroxyl or all cyclic amino acids?
- Specify that secondary structure is "3-class"
- On your table, you should clearly write the three test sets used for remote homology.
- The authors may want to see "Self-Supervised Contrastive Learning of Protein Representations By Mutual Information Maximization" by Lu, et al. which came out around the time of the ICLR deadline.
- It would be helpful to summarize on a table which data augmentations worked best for each method

Overall, this is an underexplored area, but is incremental to previous work. If the authors had shown better results, I would have increased my rating. However, the lack of rigor, novelty, nor impressive results means this paper does not meet the high bar for an impactful paper at ICLR.

---

> ### Author Response · Authors · 2020-11-21
> **RE: reviewer1 (part2)**
>
> **3. The data augmentation is done on the validation set.** *This means there is no longer a validation set! Their model performs better than the ones presented by Rao, et al. but that could be because additional data was incorporated here. The authors include a baseline where they continued training Rao, et al. on the validation set, but it would be better practice to augment data from the training set.*
>
> **ANSWER:** We would like to respectfully note that this comment is not consistent.  This can be seen clearly in our Figure 4, which does a rigorous study with varying quantities of data augmentation, in all cases following our exact training procedure, where we compare to a second internal baseline following our re-training methods with no data augmentations used.  Furthermore, Table 1 provides both a comparison to the TAPE baseline as well as this second intern baseline. Indeed, we do see an improvement in comparison to TAPE in this second baseline, but the data augmentations further improve over that, which specifically answers your question.  We have made a significant effort to account for rigorous comparisons, including the effects of the differences you have called out.  We recommend rereading the results of these two figures and the resulting discussion to see if you feel our approach is still controversial.
>
> Furthermore, the TAPE testing sets are chosen in a way such that the samples in the testing sets are out of distribution samples. As a result, it is not a conflict when training is performed with in-distribution samples (validation set) and testing is performed with out-of-distribution samples (testing sets).
>
> **4. Contact Prediction and CASP 12** *The authors do not evaluate on contact prediction because CASP12 "has an incomplete test set due to data embargoes." However, CASP12 ended 4 years ago and the data is now available. In fact, the authors use CASP12 for secondary structure prediction, so clearly they have access to the data. Furthermore, the authors could consider additional structural sets, such as structural holdout at family level; temporal holdout to CASP12/CASP13; other at the very least a sequence based split.*
>
> **ANSWER:** As you indicate, the absence of this task is due to caveats around the contact prediction dataset given by the TAPE authors, which is prepared based on the ProteinNet repo (https://github.com/aqlaboratory/proteinnet) that still indicates substantial quantities of high-quality structures embargoed from this version of the testing set.  To avoid the unknown effects of the embargoed structures on the contact prediction assessment, we made a judgment call to remove it and perform no experiments on this task.  This was a hard decision for us, because we agree that we could have just included them in our study, likely with little change to our conclusions given the consistency of the local random sub-sampling augmentation on the other structural tasks.  However, as you pointed out by mentioning the CASP competitions, we did not feel that including this task would even provide a fair comparison to the latest round of high-performing CASP12 ML models, such as AlphaFold and RaptorX, which were evaluated on the non-embargoed version of these data.  We thought it cleanest to just omit it.
> Additionally, we point out this paper’s major focus is on data augmentation and contrastive learning; we feel that our experiments on the remaining 4 TAPE tasks are sufficient to convey our conclusions, especially given that we have shown results on 2 other tasks related to protein structure.
>
>
>
> **5. Contribution:** *Overall, this is an underexplored area, but is incremental to previous work. If the authors had shown better results, I would have increased my rating. However, the lack of rigor, novelty, nor impressive results means this paper does not meet the high bar for an impactful paper at ICLR.*
>
> While we disagree with your conclusion that this paper is incremental and lacks rigor, we certainly thank you for the detailed comments and thorough reading of our paper.  However, we really want to emphasize that the goal of this paper is not strictly to find record-breaking SOTA results, which we interpret to be the point of your “nor impressive results” determination. Indeed, we strongly contest that our results are not impressive, as we have convincingly beaten both public (TAPE) and internal benchmarks in an apples-to-apples fashion, as can be most clearly seen in Tables 1 and 2.  We have made choices to promote reproducibility and a rigorous comparison to an existing, publicly available, and high-impact baseline over chasing SOTA (although we did find SOTA results in some tasks).  As this is an empirical paper in a newly growing ML field that studies the effects of data transformations for any type of semi-supervised training, we wish to stay relevant beyond the short period of time over which we may be SOTA or over which the papers that we depend on for comparison are SOTA.

---

> ### Author Response · Authors · 2020-11-21
> **RE: reviewer1 (part1)**
>
> Thank you for providing this detailed comments! Please see the responses below.
>
> **1. SOTA model:** *All evaluation was performed on the TAPE model, which was released >1 year ago. Since then, there have multiple models released that have much better performance (ProTrans from Elnaggar, et al.; ESM-1 from Rives, et al.)*
>
> **ANSWER:** This is an important comment, as the underlying question here is “why did we choose TAPE to start with, rather than something else?”.  Our clearest reason is that the TAPE authors made significant effort to promote re-use of their work and the ability to replicate their results.  Starting with their models, with exactly their data, and training with the same model hyperparameters allowed us to consistently focus on the effects of contrastive learning and their required data augmentations without distractions.  The TAPE models also rely on the highly influential BERT-style methodology, where the other approaches are modifications to this baseline with varying improvements.
>
> We also question whether pursuing SOTA with this type of data augmentation approach in this burgeoning field is really worthwhile, given that the point of our paper is that one should use data augmentations and contrastive training methods, regardless of your underlying model/architecture choices.  Entangling these results with a mix of models using different methods and with their own unique training procedures was something we wished to avoid.  Our other reasons are more pragmatic: SimCLR-v2 and ProTrans were not publicly available until very recently, making it impossible to use them in this study.  From the SOTA model perspective, we find that ProtTrans only provides improved results with secondary structure. Again, we wish to provide insights on the protein augmentations themselves, not over-indexing on hitting SOTA on individual sub-tasks. In addition, compared to the ESM-1 model, we also find outperformance of the stability task and similar performance in the Fluorescence with our model, which indicates this augmentation-based approach based only on TAPE still yields SOTA results on some of the protein tasks; and our improvements from just the TAPE baseline are roughly comparable to the amount of improvement seen by the other authors’ modeling choices.
>
>
> **2. Recent Works:** *Related work is incomplete. For example, it says SimCLR is the current SOTA contrastive learning technique. There is already SimCLR-v2 that should be cited, among others. Even though the authors use the models from Rao, et al. 2019, the authors should also cite Alley, et al. 2019, Heinzinger et al. 2019, and Rives, et al. 2019 which were first to introduce semi-supervised learning for protein language models.*
>
> **ANSWER:** We did cite and discuss Rives 2019, but we will add the various citations to Alley et al 2019, Heinzinger et al 2019, and SimCLR-v2.

---

### Official Review · AnonReviewer4 · 2020-10-30
**Strong paper that advances data augmentation regime for protein sequence data**

**Rating:** 9
**Confidence:** 4

**Review:**

Summary:
The paper explores the impact of different types of data augmentations for protein sequence data, and does a thorough benchmark analysis on them. The authors used a pre-trained transformer model, fine tuned the model on augmented data using two approaches, namely, contrastive learning and masked token prediction. This finetuned model was evaluated with an added linear layer on a range of tasks.

Overall, I vote for strong acceptance as I am genuinely excited about the paper. My reasons are below-

Pros:
1. There are no known good augmentations for protein sequences. This is probably the first paper that tries a wide range of data augmentations for protein sequences, and tests these augmentations on a wide range of benchmarked tasks.
2. The design of the data augmentations is creative. Specially, random dictionary and random alanine replacement augmentations seem to improve the downstream performances quite consistently.
3. The paper has done comprehensive experiments showing most possible configurations of the augmentations and their combination with contrastive learning and mask token prediction.

The paper is so comprehensive in its experiments in the scope of a conference that I can’t really think about any strong cons. I would just ask the authors if there are any plans to bring these augmentations to a library so that the community can start trying them rather than implementing the augmentations themselves? Thanks for this wonderful paper.


UPDATE on 12/7/2020:
I read the other reviews and the authors’ replies.
I hold onto my score and recommend the paper to be accepted. Here are some points I would like to highlight.

1. Anonreviewer1 suggests the work is incremental, yet there’s no citation to previous work on protein data augmentation. This is in fact the first major work tackling this issue.
2. Anonreviewer1 points out the authors did not use a SOTA model. This is true, but the model they are using is little more than 1 year old. Moreover, while I agree with Anonreviewer1 that different models need to be used to see how generalizable the data augmentations are, I don’t see this as a reason to reject a paper that is tackling an orthogonal issue of what modes of protein data augmentations work.
3. Both anonreviewer1 and anonreviewer3 are pointing out that different augmentations seem to have worked best for different tasks, and hence the results are not strong or useful. But how would we know this if no one had even tried it? The authors found the results as such and that’s an important contribution to the community's knowledge.
4. I am not seeing the problem of data augmentation done on the validation set that anonreviewer1 is suggesting. The test set results should not be affected in a corrupt way in this case.
5. I disagree with anonreviewer3’s comment that the augmentations are with unreasonable intuitions. I don’t think that’s an objective reason to reject a paper.
6. I agree with other reviewers that the contact prediction task should have been evaluated with whatever dataset that was available. Even then, the other 4 tasks in my opinion are enough as a first attempt at this area.

Overall, investigating the protein data augmentation regime is something that is not fancy like building a new model or beating the latest SOTA yet a necessary task for which we should thank the authors. There are of course problems that will indeed need to be addressed in further work but there are also important insights in the paper.

---

> ### Author Response · Authors · 2020-11-21
> **RE: Reviewer4**
>
> Thanks a lot for the appreciation of this work!  We also agree with you that there are no known good augmentations for protein sequences, especially as it relates to straightforward manipulations to the primary sequence.  The real value of this work in our opinion is to create debate around how much and what type of data augmentation we are willing to use, especially if those augmentations do not perfectly capture all of the possible invariances of the protein model.  We have shown that even if the a priori value of the data augmentations is ambiguous, proceeding empirically can result in improved performance.

---

### Decision · Program_Chairs · 2021-01-07
**Final Decision**

**Decision:**

Reject

**Comment:**

This paper tests out some straightforward data augmentation strategies on the protein inputs to the transformer used in the TAPE paper.  Overall, there is insufficient intellectual merit to warrant publication at ICLR. As a side-note, the quality of the manuscript in terms of scholarliness of presentation was overall lacking.